Corrected: Publisher correction

# Studying light-harvesting models with superconducting circuits

Anton Potočnik [1], Arno Bargerbos[1], Florian A.Y.N. Schröder [2], Saeed A. Khan[3], Michele C. Collodo[1], Simone Gasparinetti[1], Yves Salathé[1], Celestino Creatore[1], Christopher Eichler[1], Hakan E. Türeci[3], Alex W. Chin[2] & Andreas Wallraff [1]

The process of photosynthesis, the main source of energy in the living world, converts sunlight into chemical energy. The high efficiency of this process is believed to be enabled by an interplay between the quantum nature of molecular structures in photosynthetic complexes and their interaction with the environment. Investigating these effects in biological samples is challenging due to their complex and disordered structure. Here we experimentally demonstrate a technique for studying photosynthetic models based on superconducting quantum circuits, which complements existing experimental, theoretical, and computational approaches. We demonstrate a high degree of freedom in design and experimental control of our approach based on a simplified three-site model of a pigment protein complex with realistic parameters scaled down in energy by a factor of $10^5$. We show that the excitation transport between quantum-coherent sites disordered in energy can be enabled through the interaction with environmental noise. We also show that the efficiency of the process is maximized for structured noise resembling intramolecular phononic environments found in photosynthetic complexes.

[1] Department of Physics, ETH Zurich, CH-8093 Zürich, Switzerland. [2] Cavendish Laboratory, University of Cambridge, J. J. Thomson Avenue, Cambridge CB3 0HE, UK. [3] Department of Electrical Engineering, Princeton University, Princeton, NJ 08544, USA. These authors contributed equally: Anton Potočnik, Arno Bargerbos. Correspondence and requests for materials should be addressed to A. Potočnik (email: anton.potocnik@phys.ethz.ch)

I t is well accepted that the microscopic properties of all matter being composed of atoms and molecules are governed by the laws of quantum physics. At macroscopic scales, however, coherent quantum phenomena are frequently suppressed by the interaction with the environment. An intensely studied open question is, whether quantum mechanics plays an important functional role in biological processes. Examples of such processes are magnetoreception in birds, olfaction, and light harvesting, all studied in a field referred to as quantum biology[1–3]. In particular, quantum-coherent effects were observed in photosynthetic complexes by 2D electron spectroscopy at near-ambient conditions[4–6], which stimulated both experimental and theoretical work on light harvesting.

In a photosynthetic process, light is captured in a molecular complex acting as an antenna. The created excitation is then relayed toward a reaction center through a network of chlorophyll molecules forming pigment protein complexes, such as the well studied Fenna–Matthews–Olson (FMO) complex[1,2]. At the reaction center the excitation enables the synthesis of energy-rich molecules, e.g., adenosine triphosphate, relevant for supplying chemical energy throughout an organism. An excitation of an individual chlorophyll molecule is carried by a single chromophore whose highest occupied and lowest unoccupied molecular orbital can be approximated as a two-level system. The chlorophyll molecules form the sites of a network through which the individual excitations are transported. The energy levels of the individual sites and the coupling between the sites are affected by both static and dynamic disorder, which in uniform systems suppresses energy transfer between sites.

High-efficiency energy transport between disordered sites is suggested to be enabled through the interaction of the individual sites with vibrational modes of the protein scaffold into which the chlorophylls are embedded. A number of theoretical models have been developed to put this mechanism of efficient energy transport onto a solid footing[7–11]. In particular, it has been suggested that interactions between the energy levels of the chlorophyll molecules and the highly structured phononic environment of the protein scaffold enhance directed excitation transport[12–15].

The direct verification of these models is challenging due to the intricate structure and the limited control obtainable over photosynthetic complexes. Despite a number of theoretical studies, noise-assisted energy transport (NAT) in biological systems has so far only been phenomenologically investigated on simple model systems with limited control over its parameters. Energy transport between two molecules placed on a substrate was studied with scanning tunneling microscopy[16], using classical optics disorder was shown to break destructive interference and increase optical transmission[17–19]; similarly, disorder in the coupling parameter was shown to increase energy flow using classical electronic circuits[20] and in genetically engineered molecular systems energy transport was controlled by adjusting interchromophoric distances[21,22]. Recently, a programmable nanophotonic processor was used to study the transport properties in disordered systems[23].

In this work we demonstrate the use of superconducting quantum circuits[24,25] to test models describing important aspects of photosynthesis, such as photon absorption and noise assisted excitation transport, with unprecedented control in an engineered quantum system. We realize a small network of coherently coupled two-level systems with in situ tunable parameters interacting with an engineered environment, inspired by the proposal of Mostame et al.[26,27]. Superconducting circuits are particularly well suited for this task, since versatile devices can be realized with a high degree of accuracy, and can be controlled and probed experimentally using well-developed techniques[28]. Based on recent developments in circuit design, control, and measurement

with efforts aimed at realizing circuits for quantum information processing[29–31], it seems likely that our approach could be extended to study more complex quantum networks. We experimentally demonstrate energy transport assisted by structured and unstructured environmental noise for coherent and incoherent excitation and show that its efficiency can approach unity. We also observe static coherences, even under incoherent excitation, and demonstrate good understanding of the full system dynamics.

## Results

**Sample and spectroscopic characterization.** We implement a simplified model of a pigment protein complex consisting of three coupled chlorophyll molecules, labeled $Q_{1,2,3}$ in Fig. 1a. The corresponding Hamiltonian is described in Supplementary Note 1. This is the smallest system which incorporates all relevant elements, such as excitation trapping, energy mismatch, excitation delocalization, and dark and bright states, necessary for studying noise-assisted transport[1,9,11,13–15]. Although current technology allows building larger systems capable of investigating, for example the full FMO complex with eight sites, we are convinced that it is a necessary first step to explore this novel approach on a simple model system. We realize two-level systems with individually tunable transition frequencies as transmon qubits[32] in a superconducting circuit (Fig. 1b). The dipole–dipole coupling between molecules $Q_1$ and $Q_2$ forms symmetric and antisymmetric, bright $|b\rangle$ and dark $|d\rangle$ state superpositions of the individual qubit excited states $|q_1\rangle$ and $|q_2\rangle$[33–35]. We realize the dipole–dipole interaction by direct capacitive coupling between qubits $Q_1$ and $Q_2$ (Fig. 1b). We excite the bright state $|b\rangle$ through an open waveguide to which the two transmon qubits are coupled with equal strength[35] modeling the excitation of the antenna part of the photosynthetic complex with photons propagating in free space. A third molecule $Q_3$, coupled to $Q_2$, acts as a trap for the excitation which is subsequently extracted by transfer to the reaction center. In our circuit, this trapping site is realized as a third qubit which extracts the excitation from the system through its Purcell-like coupling to a transmission line resonator effectively acting as the reaction center. We model the interaction of molecule $Q_2$ with the environmental vibrational modes of the protein scaffold as fluctuations of its transition frequency. These fluctuations are induced by local magnetic fields acting on $Q_2$. While noise can be applied to all qubits, we chose to study local noise, such as the one induced by a local environment of a pigment protein. When applying noise to several sites, we could study the effects of correlations in the noise have on the process.

We demonstrate a high degree of tunability of system parameters in a measurement of the frequency-dependent transmission coefficient $|t_{21}(\omega)|$, through the waveguide (Fig. 2a), keeping the transition frequency of $Q_3$ fixed at $\omega_3/2\pi = 6.198$ GHz and linearly sweeping the transition frequencies of $Q_1$ and $Q_2$ maintaining $\omega_1 = \omega_2$. We tune the qubit transition frequencies by magnetic fields applied using a coil and two flux lines shorted close to the SQUID loops of transmon qubits $Q_1$ and $Q_2$ (see Supplementary Note 1). In this measurement, we observe that $Q_1$ and $Q_2$ form bright and dark states ($|b\rangle$, $|d\rangle$) with frequencies $\omega_b$ and $\omega_d$ separated by $2J_{12}/2\pi = 173.4$ MHz (see Supplementary Note 2). The bright state linewidth $\gamma_b/2\pi = 12.4$ MHz is consistent with the sum of the individual qubit radiative linewidths $\gamma_1/2\pi = 7.39$ MHz and $\gamma_2/2\pi = 6.57$ MHz dominated by the coupling to the waveguide. This indicates superradiance of the coupled two-qubit system[35,36]. The subradiant dark state $|d\rangle$ has a narrow linewidth of $\gamma_d/2\pi = 0.29$ MHz limited by the residual asymmetry in $Q_1$ and $Q_2$ parameters (see Supplementary

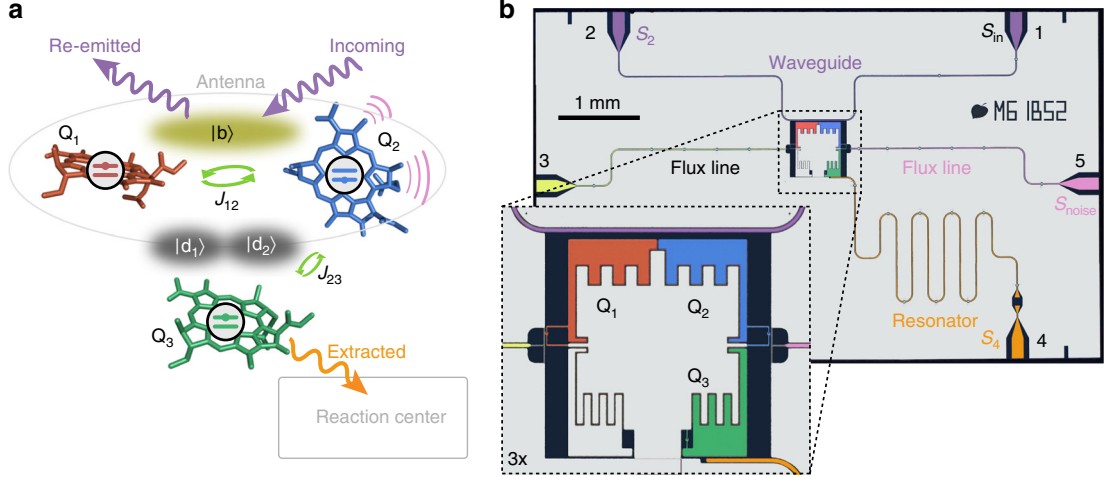

**Fig. 1** Model system for light harvesting and its superconducting circuit realization. **a** Schematic of three coupled chlorophyll molecules hosting chromophores modeled as three qubits $Q_{1,2,3}$ (red, blue, and green). The strongly coupled $Q_1$ and $Q_2$ hybridize with coupling strength $J_{12}$ into a bright state $|b\rangle$ (olive cloud) and dark states (dark gray clouds). $Q_2$ is coupled to $Q_3$ with a coupling strength $J_{23}$ which forms the dark hybridized states $|d_1\rangle$ and $|d_2\rangle$. The incident, the re-emitted, and the harvested radiation are indicated by arrows. Pink lines indicate that $Q_2$ is subjected to environmental noise. **b** False color micrograph of the superconducting circuit. Transmon qubits $Q_1$ and $Q_2$ (red, blue) are capacitively coupled to a transmission line (purple) and $Q_3$ (green) is capacitively coupled to a high-emission-rate resonator (orange). Excitation radiation ($S_{in}$) is applied through port 1 of the transmission line. The re-emitted ($S_2$) and extracted radiation ($S_4$) are detected at port 2 and 4, respectively. Low-frequency noise with an engineered spectral density, modeling the environment ($S_{noise}$), is applied to $Q_2$ via the flux line at port 5

Note 1) with a bright to dark state linewidth ratio of $\gamma_b/\gamma_d = 43$. Dark states have been suggested to improve the efficiency of biologically inspired photocells by protecting the excitation from re-emission[14,37]. For $\omega_1/2\pi = \omega_2/2\pi = 6.285$ GHz (solid vertical line in Fig. 2a) the dark state $|d\rangle$ coherently hybridizes with $|q_3\rangle$ forming a doublet $|d_1\rangle$ and $|d_2\rangle$ split by $2J_{d3}/2\pi = 37$ MHz consistent with the individual qubit couplings (see Fig. 2b and Supplementary Notes 1 and 2). We detune $\omega_3$ from the fixed frequency $\lambda/2$ resonator state $|r\rangle$ at $\omega_r/2\pi = 6.00$ GHz by $\Delta_{3r} = \omega_3 - \omega_r$. This sets the radiative Purcell decay rate[38] $\gamma_{Pur}/2\pi = 20$MHz of $Q_3$ (see Supplementary Note 3) effectively modeling the energy extraction rate at the reaction center. As desired, all relevant microwave frequency system parameters are consistently scaled by a factor of $\sim 10^5$ relative to the optical-frequency energy scales of the FMO complex[26].

**Excitation transfer with uniform white noise.** To study energy transfer through the circuit, we tune $Q_1$ and $Q_2$ into resonance to form bright and dark states at frequencies $\omega_b/2\pi = \nu_b = 6.371$ GHz and $\omega_d/2\pi = 6.198$ GHz. Qubit $Q_3$ is tuned into resonance with the dark state $\omega_3 = \omega_d$ creating two resonances at $\omega_{d1}/2\pi = \nu_{d1} = 6.179$ GHz and $\omega_{d2}/2\pi = \nu_{d2} = 6.216$ GHz. We coherently excite the bright state $|b\rangle$ through port 1 of the device with a continuous tone at frequency $\omega_b$ and amplitude corresponding to a bright state Rabi frequency of $\Omega_R/2\pi = 14$ MHz (see Supplementary Note 4). We measure the power spectral density (PSD) $S_2(\omega)$ of the photons scattered along the waveguide into port 2 of the device characterizing the re-emission from the absorption site. $S_2(\omega)$ displays a narrow coherent peak at $\omega_b$ due to elastically (Rayleigh) scattered photons and a broad resonance fluorescence spectrum with a width given by $\gamma_b$ due to the inelastically scattered photons (bottom purple line in Fig. 3a). With increasing drive amplitude we observe a bright state Mollow triplet[35], see Supplementary Fig. 4. Due to the energy mismatch of the bright state $|b\rangle$ and the dark state doublet ($|d_1\rangle$, $|d_2\rangle$) no excitations are transferred to qubit $Q_3$ and thus no photons are detected at the resonator port 4, as shown by the vanishing PSD $S_4(\omega)$ at $\Phi_w^2 = 0\,pWb^2$ (bottom orange line in Fig. 3a). In our model

system, no energy is transferred from the antenna to the reaction center in the absence of environmental noise.

To engineer a broad environmental noise spectrum, such as the one generated by the combination of background thermal noise and overlapping vibrational modes of the protein scaffold present in light-harvesting systems[26], we apply white Gaussian noise to port 5 inducing frequency fluctuations in $Q_2$. The broad Markovian noise has a PSD of adjustable amplitude, constant up to a cutoff frequency of 325 MHz, characterized by its integrated flux noise power $\Phi_W^2$ at qubit $Q_2$ (see Supplementary Note 5 and Fig. 2c). We note that applying synthesized noise to $|q_2\rangle$ effectively creates a classical environment that can be described by the Haken–Strobl–Reineker model[39,40] for white noise. Applying classical, as opposed to quantum noise, offers a unique possibility to engineer noise with controllable PSD capable of creating environments that approximate those of pigment protein complexes[26] without increasing complexity of the device design.

For small applied noise powers, we observe energy transfer from the bright $|b\rangle$ to the dark state doublet $|d_1\rangle$, $|d_2\rangle$ indicated by two resonances at frequencies $\omega_{d1}$ and $\omega_{d2}$ in the detected PSD $S_4(\omega)$ (orange line at $\Phi_W^2 = 0.1\,pWb^2$ in Fig. 3a). The excitation transport is enabled by those frequency components of the noise spectrum that bridge the energy difference $2J_{12} \pm J_{d3}$ between bright $|b\rangle$ and dark states $|d_1\rangle$, $|d_2\rangle$. We have verified this aspect by reducing the bandwidth of the noise to below that energy difference in which case no energy transfer is observed. The emission linewidths, i.e., the emission rates of $|d_1\rangle$ and $|d_2\rangle$ into the resonator are determined by the Purcell decay rate $\gamma_{Pur}$ (see Supplementary Note 3). The well-resolved doublet in the detected spectrum $S_4(\omega)$ indicates that static coherences of the underlying quantum network are observable in noise induced transport. Based on the observations of the doublet, we expect beatings with frequency $2J_{d3}$ to be observable in temporally resolved measurements of the power at the extraction site.

With increasing applied noise power $\Phi_W^2$, the power spectrum $S_2(\omega)$ of the resonance fluorescence of the bright state $|b\rangle$ broadens due to the pure dephasing induced by the noise. From this measurement we determine the bright state pure

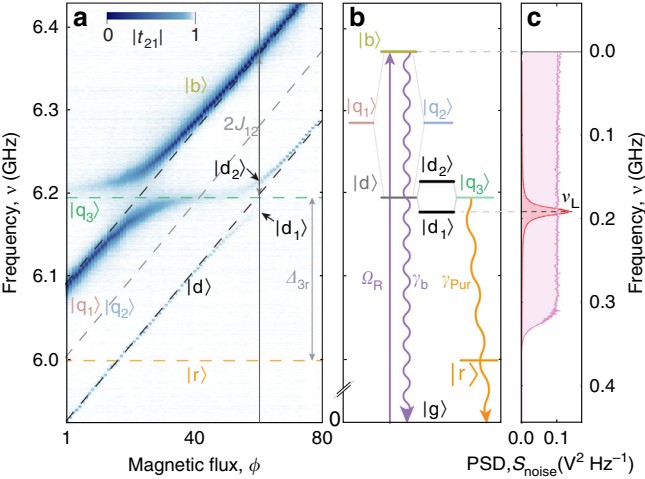

**Fig. 2** Measured spectrum, energy levels, and applied environmental noise spectra. **a** Transmission spectra $|t_{21}(\omega)|$ of the three-qubit system measured through the transmission line as a function of magnetic flux. Here and in the following, spectral features are labeled by the target state ($|q_1\rangle$, $|q_2\rangle$, $|q_3\rangle$, $|b\rangle$, $|d\rangle$, $|d_1\rangle$, $|d_2\rangle$, $|r\rangle$) reached in spectroscopic experiments from the ground state $|g\rangle$ of the system. **b** Energy-level diagram at the magnetic flux indicated by a gray vertical line in **a**. The resonant states $|q_1\rangle$ and $|q_2\rangle$ form bright $|b\rangle$ and dark $|d\rangle$ states. Furthermore, $|q_3\rangle$ is resonant with the $|d\rangle$ state forming $|d_1\rangle$ and $|d_2\rangle$ doublet. Solid downward arrows indicate decay channels; the upward arrow indicates excitation via the waveguide. **c** Measured power spectral density (PSD) $S_{\text{noise}}$ of the environmental low frequency noise applied to the flux line of $Q_2$. White noise with 325 MHz cutoff is depicted in pink and Lorentzian noise with central frequency $\nu_L$ in red

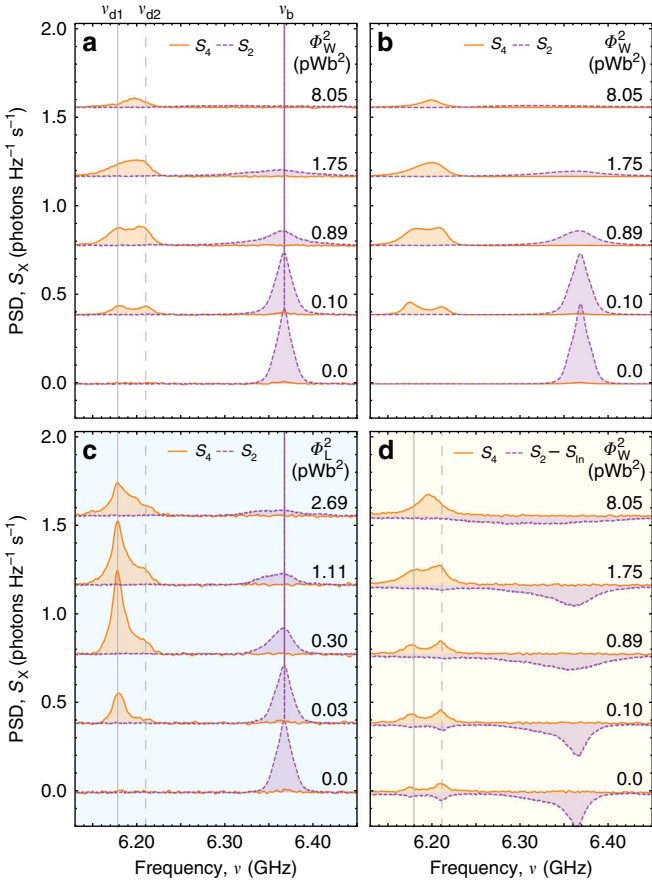

**Fig. 3** Measured power spectral densities (PSD). PSD of radiation extracted from the resonator $S_4(\omega)$ (solid orange lines) and re-emitted into the transmission line $S_2(\omega)$ (dashed purple lines) for coherent excitation as a function of **a** white noise power $\Phi_{\text{W}}^2$ and **c** Lorentzian noise power $\Phi_{\text{L}}^2$. **b** Master equation calculations of $S_4(\omega)$ and $S_2(\omega)$ for coherent excitation as a function of $\Phi_{\text{W}}^2$. **d** Measured PSD for incoherent excitation as a function of $\Phi_{\text{W}}^2$. Dashed purple lines represent a difference between measured $S_2(\omega)$ and PSD of incoherent microwave radiation $S_{\text{in}}$. PSD for different noise powers are displaced by 0.2 photons s$^{-1}$ Hz$^{-1}$

dephasing rate $\gamma_\phi^b$ in dependence on the applied white noise power $\Phi_{\text{W}}^2$ (see Supplementary Note 6). The extracted power $S_4(\omega)$ first increases with increasing noise power $\Phi_{\text{W}}^2$ while the doublet remains resolved. At noise powers above $\Phi_{\text{W}}^2 \approx 2$ pWb$^2$, the observed doublet transforms into a single resonance marking a crossover from the strong-coupling regime ($2J_{d3} \gtrsim \gamma_\phi^b$) to the weak-coupling regime ($2J_{d3} \lesssim \gamma_\phi^b$), where the remaining resonance stems from the incoherently excited $|q_3\rangle$ state (see Supplementary Note 7). Beyond this threshold, the extracted power decreases. For this simple situation of only three sites and Markovian noise, all essential features of the experimentally observed power spectra are consistent with both Lindblad master equation and Bloch–Redfield calculations (Fig. 3b and Supplementary Notes 8, 9).

Integrating the measured PSD $S_4(\omega)$ and $S_2(\omega)$ while omitting contributions from elastic (Rayleigh) scattering, we find that the total power re-emitted from the bright state into the waveguide in forward direction $P_2$ decreases monotonically as a function of applied noise power $\Phi_{\text{W}}^2$ (open purple squares in Fig. 4a). In contrast, the total power detected at the extraction site $P_4$ first increases rapidly with $\Phi_{\text{W}}^2$, exhibits a pronounced maximum and then decreases again (open orange diamonds in Fig. 4a). The increase for small dephasing rates is a consequence of noise-induced incoherent transitions between bright and dark states[14,41] allowing the system to overcome the energy mismatch.

From the integrated powers we calculate the transfer efficiency of the excitation from the absorption site to the extraction site as $\eta = P_4/(P_4 + 2P_2)$. The Factor 2 accounts for the bidirectional character of the bright state resonance fluorescence[35], i.e., equal powers are emitted in forward and backward direction, while we detect only in forward direction. The transport efficiency $\eta$ (green circles in Fig. 4a) shows a rapid increase from zero, a broad maximum of $\eta_{\text{W}}^{\text{max}} = 39\%$, and then a slow decrease with

increasing pure dephasing rate $\gamma_\phi^b$ (top axis in Fig. 4a), which are the characteristic features of noise assisted transport[8–10]. The decrease in efficiency above an optimal noise power is due to dephasing-induced population localization, also referred to as the quantum Zeno effect[8–10].

The measured integrated powers and hence the efficiency are in good agreement with results from master equation simulations (solid lines in Fig. 4a). Using rate equations (Supplementary Note 10), we show that the maximal efficiency is $\eta_{\text{W}}^{\text{max}} \approx (1 - \gamma_b/\gamma_{\text{Pur}})$ approaching 100% for $\gamma_b \ll \gamma_{\text{Pur}}$. Although small $\gamma_b$ maximizes the transfer efficiency[10], the total extracted power at optimal applied noise is proportional to $\gamma_b$. Therefore for practical light harvesting applications one may choose to maximize output power while compromising on efficiency[14,41]. Similarly, we have not chosen a smaller $\gamma_b$ in our experiment to maximize the efficiency, but opted for a larger extracted power to achieve a high signal-to-noise ratio at an acceptable integration time. Finally, we note that in our data the maximum efficiency $\eta_{\text{W}}^{\text{max}}$ occurs at the strong-to-weak coupling crossover $2J_{d3} \approx \gamma_\phi^b$. At this point, the transfer rate $\gamma_\phi^b$ between $|b\rangle$ and $|d\rangle$ is comparable to the transfer rate between $|d\rangle$ and $|q_3\rangle$, which is approximately given by $2J_{d3}^2/\gamma_\phi^b$ (see Supplementary Note 10). This experimentally demonstrates the interplay between

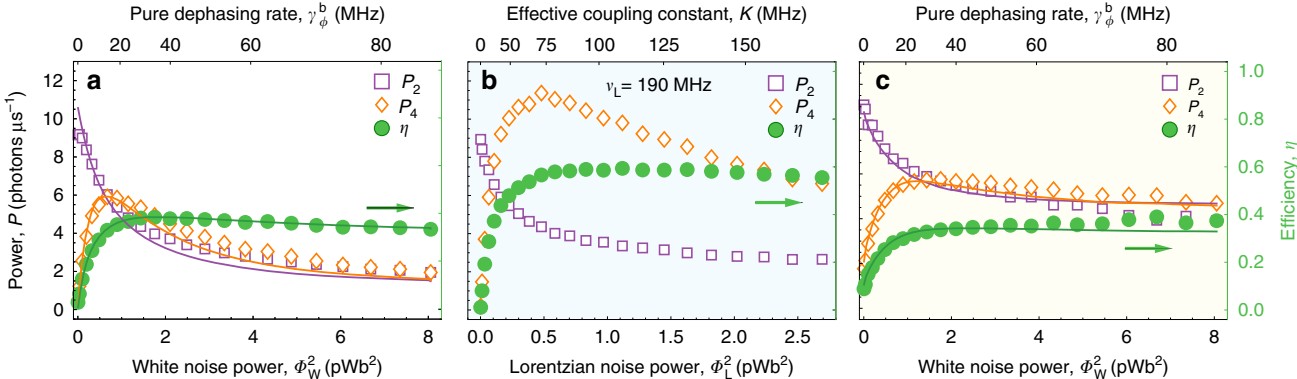

**Fig. 4** Total extracted power $P_4$, re-emitted power $P_2$, and transport efficiency $\eta$ for **a** coherent excitation and white noise environments, **b** coherent excitation and Lorentzian noise environments, and **c** incoherent excitation and white noise environments. Data are plotted as a function of white noise power $\Phi_W^2$ or equivalently bright state dephasing rate $\gamma_\phi^b$ for **a** and **c** and as a function of Lorentzian noise power $\Phi_L^2$ or equivalently effective qubit-environment coupling constant $K$ for **b**. Solid lines are results of master equation simulations (see Supplementary Note 9)

quantum coherent effects and classical dephasing enhancing excitation transport.

**Excitation transfer with lorentzian environment**. It has been conjectured that structured environmental noise, such as the one originating from long-lived vibrational modes of chlorophyll molecules in photosynthetic complexes, can further enhance the energy transfer efficiency between the disordered molecular sites of the network in a scenario known as the phonon antenna mechanism[1,12]. To demonstrate this concept, we apply environmental noise with Lorentzian PSD, characterized by its central frequency $\nu_L$, its width $\Delta\nu_L$ and amplitude (Fig. 2c), to qubit $Q_2$. We select a fixed bandwidth $\Delta\nu_L = 10$ MHz, which in good approximation corresponds to the scaled linewidth of the environmental noise expected from vibrational modes in natural photosynthetic complexes[15,26]. Since $\Delta\nu_L$ is comparable to the decay rates $\gamma_b$ and $\gamma_{Pur}$, the spectral properties of the noise effectively create a non-Markovian environment (see Supplementary Note 11). Initially, we choose the Lorentzian central frequency $\nu_L = 190$ MHz to be resonant with the $|b\rangle$ to $|d_1\rangle$ frequency difference $\Delta_{b,d1}$. The qualitative features of the measured PSD $S_2(\omega)$ and $S_4(\omega)$ and their dependence on the integrated applied noise power $\Phi_L^2$ (Fig. 3c) are comparable to the white noise case (Fig. 3a) with some distinct differences. As a direct consequence of applying Lorentzian noise resonant at $\Delta_{b,d1}$, $S_4(\omega)$ exhibits a strong resonance exactly at $\omega_{d1}$ and a weaker resonance at $\omega_{d2}$. The excitation transfer can be interpreted as a two-photon process[42] absorbing one photon from the coherent excitation field at frequency $\omega_b$ and emitting one photon into the environmental noise field at frequency $\Delta_{b,d1}$ or $\Delta_{b,d2}$. This effectively creates a transition from the joint ground state through the bright state $|b\rangle$ into the dark states $|d_1\rangle$ or $|d_2\rangle$ from which energy is extracted.

When we sweep the Lorentzian noise center frequency over a broad range from $\nu_L = 0$ to 300 MHz at weak noise power $\Phi_L^2 = 0.016$ pWb$^2$, we observe two well-resolved maxima in transferred power $P_4$ when the noise is resonant with the bright to dark state frequency differences $\Delta_{b,d1}$ and $\Delta_{b,d2}$ (Supplementary Fig. 8). In contrast, no energy transfer is observed when the noise is far detuned. Both observations clearly demonstrate the strong sensitivity of the energy transfer on the spectral properties of the environmental noise, and indicate that the noise-assisted transport is enabled by a narrow part of the noise PSD that matches the energy gap otherwise blocking the energy transfer in the system. This observation is consistent with stochastically averaged master equation simulations (see Supplementary Fig. 9 and Supplementary Note 12). We note that despite applying

classical noise the dynamics induced by Lorentzian environment cannot be simulated with the HSR approach or even its extension for colored noise[43,44], due to its strong non-Markovian character. Therefore, a full quantum numerical simulation is required at the cost of significant computational effort.

For Lorentzian environmental noise, the integrated detected powers also display the characteristic properties of noise-assisted transport as a function of $\Phi_L^2$ (Fig. 4b) as discussed before for white noise (Fig. 4a). However, we note that the extracted power $P_4$ is almost twice as large at the same bright state excitation amplitude (Fig. 4a, b) leading to increased maximal efficiency $\eta_L^{max} = 58\%$. This indicates that structured environmental noise matching internal energy differences of the quantum network indeed enhances the efficiency of the energy transfer. Estimating the effective qubit-environment coupling constant $K$ (see Fig. 4b and Supplementary Note 13), we observe that the maximum in efficiency coincides with $K/2\pi \approx 100$ MHz, which is comparable to the inter-qubit coupling constant $J_{12}$. This demonstrates that strong coupling between qubit and phononic modes is required to achieve maximal energy transport. Such a situation has been suggested to enhance the transport in cyanobacterial light-harvesting proteins, allophycocyanin, and C-phycocyanin[45].

We have also applied a coherent tone with controlled frequency $\nu_c$ and amplitude to qubit $Q_2$ through the environmental channel (port 5) (see Supplementary Note 14), and observe even larger extracted powers at the same bright state $|b\rangle$ input field amplitude and near unit transfer efficiency (see Supplementary Fig. 10c). While this case does not occur in natural light harvesting since the environment cannot be fully coherent, it represents an interesting limiting case. In natural light-harvesting systems exposed to incoherent environments, the highest transfer efficiency is realized for structured environmental noise with a narrow spectral density peaked at the frequency of the internal energy mismatch between the sites of the network. Any excess spectral width of the environmental noise leads to additional dephasing (see Supplementary Note 11), which in turn reduces the absorption and energy transfer efficiency. These aspects are clearly demonstrated by the presented set of experiments comparing energy transfer with white and Lorentzian noise spectral density.

**Incoherent excitation**. In the final set of experiments we excite the qubit system with incoherent microwave radiation to mimic excitation of biological pigment protein complexes with sunlight. We engineer 0.95 GHz broad incoherent microwave radiation centered at $\omega_B$ that spans over all qubit transition frequencies (see

Supplementary Note [15]). The incoherent microwave power integrated over the bright state spectrum was adjusted to be equal to the drive power used for the case of coherent excitation. In this experiment we study the transport of incoherently created excitations as a function of applied white noise power.

Since it is not possible to distinguish between the broad incoming incoherent and the re-emitted radiation at port 2, we plot in Fig. [3]d, the difference between detected power spectrum $S_2$ at port 2 and the separately measured incoherent radiation spectrum $S_{In}$. The difference spectrum corresponds to the sum of the absorbed and the re-emitted spectrum $\tilde{S}_2$.

When increasing the applied white noise power, we observe that $S_4(\omega)$ and $[S_2(\omega) - S_{In}(\omega)]$ show general features (Fig. [3]d) similar to the ones observed for coherent excitation. However, in case of incoherent excitation, a finite power ($P_4 = 0.9$ photons $\mu s^{-1}$) is extracted at port 4 even in the absence of applied environmental white noise ($\Phi_W^2 = 0$). The observed extracted power is a result of direct excitation of the dark state, due to its finite coupling to the open waveguide. When the dark state is not completely dark, simultaneous incoherent excitation of bright and dark states reduces coherence between $Q_1$ and $Q_2$ and therefore increases dephasing of the system. Existence of dark states in photosynthetic complexes can therefore help protect the system against dephasing induced by incoherent excitation. The observation of the $|d_1\rangle$, $|d_2\rangle$ doublet in $S_2(\omega)$ (Fig. [3]d) demonstrates that static coherences can be observed for incoherent excitation, i.e., even in the absence of coherent sources, as long as the coherent coupling between the sites is larger than the total dephasing.

The maximum of the measured efficiency $\eta_{W,inc.}^{max} = 38\%$ is smaller, but comparable to the case of coherent excitation with the maximum shifted toward higher applied environmental white noise powers. Comparable efficiencies are consistent with rate equation descriptions, where the efficiency is independent of the spectral and coherence properties of the excitation. On the other hand, the extracted power $P_4$ is by more than a factor 2 larger at the highest environmental noise power ($\Phi_W^2 = 8.05$ pWb$^2$) compared to the coherent excitation case. This indicates that absorption of the incoherent photons is not as strongly affected by environmental dephasing as for coherent excitation, due to a persistent overlap between broadened bright state spectrum and spectrum of the incoherent irradiation. Similar conclusion can be made for energy transport induced by Lorentzian noise for incoherent excitation (see Supplementary Note [15]).

## Discussion

In a proof of concept experiment we studied models of photosynthetic processes using superconducting quantum circuits. With a system of three coupled qubits we demonstrated how the interplay of quantum coherence and environmental interactions affects energy transport in a system with excellent control achievable over all relevant parameters. We expect this approach to be extensible to study other relevant aspects of light harvesting, such as time-resolved dynamics of the coherent excitation transfer; the role of quantum environments realizable in electronic circuit models as low frequency quantum harmonic oscillators[14,26]; and scaling to systems with a larger number of coherent sites such as the FMO complex. Furthermore, we expect similar approaches to be applicable not only to study light-harvesting processes but also other interesting aspects of quantum biology such as the sense of smell in animals, humans, and magnetoreception in birds[1,2]. It could also be interesting to evaluate the potential of the techniques presented here to model processes in quantum chemistry and search for potential future applications of related methods to support, for example, the

design of catalysts, e.g., for nitrogen fixation, or biomolecular compounds for drug development.

**Data availability**. The authors declare that the data supporting the findings of this study are available within the article and its Supplementary Information files or from the corresponding authors on reasonable request.

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

## Acknowledgements

We are grateful to G. Blatter and D. Vion for helpful feedback on the manuscript and T. Walter and P. Kurpiers for valuable discussions. Work of H.E.T. and S.K. was supported by the US Department of Energy, Office of Basic Energy Sciences, Division of Materials Sciences, and Engineering, under Award No. DE-SC0016011. A.W.C. and F.A.Y.N.S. acknowledge support from the Winton Programme for the Physics of Sustainability. F.A. Y.N.S. also acknowledges support by the Engineering and Physical Sciences Research Council (EPSRC). Work of A.P., A.B., M.C., S.G., Y.S., C.E. and A.W. was supported by ETH Zürich.

## Author contributions

A.P., A.B. and M.C.C. designed the sample, performed the experiment, and analyzed the data. S.G. fabricated the sample. The FPGA firmware was implemented by Y.S.; F.A.Y.N. S., C.C. and A.W.C. performed numerical simulations with the uniform environment and contributed to the experimental set-up. S.A.K. and H.E.T. performed numerical simulations with the structured environment. A.P. and A.W. co-wrote the manuscript. C.E. commented on the manuscript. All authors contributed to the manuscript preparation. The project was led by A.P. and A.W.

## Additional information

**Competing interests:** The authors declare no competing interests.

