## [Peer Review File · Nature Communications]

Reviewers' comments:

Reviewer #2 (Remarks to the Author):

In this revised manuscript titled "Studying Light-Harvesting Models with Superconducting Circuits," I think the authors have properly addressed most of my technical comments.

However, I am still concerned about whether the presented work would be capable of advancing knowledge and understanding regarding photosynthetic light harvesting, although the authors claim "Here we experimentally demonstrate a new approach for studying photosynthetic models based on superconducting circuits ... the unprecedented versatility and control of our method ..."

Indeed, I totally agree to the authors' reply,

1) The great potential of the experiment using rapidly developing technology of superconducting circuits,

2) The importance of studying a smaller well-controlled system that can be completely characterized in its simplest form.

However, I am not sure how the authors could handle and emulate heterogeneity in site energies, site-environment, site-vibration interaction in disordered and complex many-site systems such as photosynthetic light harvesting proteins. In this sense, it is tough for me to understand the meaning of "the unprecedented versatility" of the authors' experiment.

Advantages of the presented approach over the conventional theoretical and computational approach should be clearly discussed.

Reviewer #3 (Remarks to the Author):

The authors have provided a thorough response to the referee questions and, importantly, have made changes to the manuscript that better explain the context of the research as well as the achievements/shortcomings of the current experiment.

I recommend publication in Nature Communications.

Reply to Reviewers Comments

In this document we reply to all of the comments from the three reviewers in detail. Our response is written in blue.

Reviewers' comments:

Reviewer #2 (Remarks to the Author):

In this revised manuscript titled "Studying Light-Harvesting Models with Superconducting Circuits," I think the authors have properly addressed most of my technical comments.

We are grateful that the referee accepted most of our comments.

However, I am still concerned about whether the presented work would be capable of advancing knowledge and understanding regarding photosynthetic light harvesting, although the authors claim "Here we experimentally demonstrate a new approach for studying photosynthetic models based on superconducting circuits ... the unprecedented versatility and control of our method ..."

Indeed, I totally agree to the authors' reply,

- 1) The great potential of the experiment using rapidly developing technology of superconducting circuits,
- 2) The importance of studying a smaller well-controlled system that can be completely characterized in its simplest form.

However, I am not sure how the authors could handle and emulate heterogeneity in site energies, site-environment, site-vibration interaction in disordered and complex many-site systems such as photosynthetic light harvesting proteins. In this sense, it is tough for me to understand the meaning of "the unprecedented versatility" of the authors' experiment.

Advantages of the presented approach over the conventional theoretical and computational approach should be clearly discussed.

The work we present in our manuscript aims at complementing existing work in that well developed area of research with a new approach based on performing analog simulations with a superconducting circuit based model system. This particular system allows for a high degree of control in both designing circuits and controlling and measuring their properties.

While requiring significant effort, we are confident that we will be able to extend the capabilities of the current circuit consisting of three qubits, one resonator, one waveguide and two flux lines to create classical fields in the environment to more complex ones. It seems likely that systems

with up to 10 sites can be explored within a year or two and systems with several tens of sites within 5 years. In addition, quantum environments can be realized by coupling low frequency tunable oscillators to the sites in a controllable fashion. The ability to design, control and measure such circuits will develop in parallel with efforts aimed at realizing related circuits for quantum information processing. For recent developments, see for example [Barends et al., Nature 534, 222 (2016). Song et al. PRL 119, 180511 (2017). Otterbach et al. arXiv:1712.05771 (2017).].

To address the concerns of the referee, we have modified part of the abstract to read:

“Here we experimentally demonstrate a new technique based on analog simulations with superconducting quantum circuits which we believe complements existing experimental, theoretical and computational approaches for studying photosynthetic models. In particular, we demonstrate a high degree of freedom in design and experimental control of our approach based on a simplified three-site model of a pigment protein complex with realistic parameters scaled down in energy by a factor of 10^5 .”

Following referee’s suggestion we have added the following discussion on the prospects of our approach at the end of the fourth paragraph of the manuscript:

“Based on recent developments in circuit design, control and measurement with efforts aimed at realizing circuits for quantum information processing [Barends et al., Nature 534, 222 (2016). Song et al. PRL 119, 180511 (2017). Otterbach et al. arXiv:1712.05771 (2017)], it seems likely that our approach could be extended to study more complex quantum networks.”

Reviewer #3 (Remarks to the Author):

The authors have provided a thorough response to the referee questions and, importantly, have made changes to the manuscript that better explain the context of the research as well as the achievements/shortcomings of the current experiment.

I recommend publication in Nature Communications.

We thank the referee for recognizing that our manuscript is suitable for publication.